# Subject validation of reusable N95 stop-gap filtering facepiece respirators in COVID-19 pandemic

**William C. K. Ng**[1,2,3☯]*, **Arnaud Romeo Mbadjeu Hondjeu**[1☯], **Andrew Syrett**[1☯],
**Rebecca Caragata**[1‡], **Dmitry Rozenberg**[4,5‡], **Zixuan Xiao**[6‡], **Vahid Anwari**[7‡],
**Jessica Trac**[5‡], **Azad Mashari**[1,2☯]

**1** Department of Anesthesiology and Pain Management, Toronto General Hospital, University Health Network, Toronto, Ontario, Canada, **2** Department of Anesthesiology and Pain Medicine, University of Toronto, Toronto, Ontario, Canada, **3** Department of Anesthesiology and Pain Management, Cardiac Division, Hospital for Sick Children, Toronto, Ontario, Canada, **4** Division of Respirology and Lung Transplantation, Toronto General Hospital Research Institute, University Health Network, Toronto, Ontario, Canada, **5** Department of Medicine, University of Toronto, Toronto, Ontario, Canada, **6** Faculty of Medicine, University of Alberta, Edmonton, Alberta, Canada, **7** Joint Department of Medical Imaging, Toronto General Hospital, University Health Network, Toronto, Ontario, Canada

☯ These authors contributed equally to this work.
‡ These authors also contributed equally to this work.
* DrWilliam.Ng@mail.utoronto.ca

**Data Availability Statement:** All relevant data are within the manuscript and its Supporting information files.

## Abstract

### Introduction

The COVID-19 pandemic has unveiled widespread shortages of personal protective equipment including N95 respirators. Several centers are developing reusable stop-gap respirators as alternatives to disposable N95 respirators during public health emergencies, using techniques such as 3D-printing, silicone moulding and plastic extrusion. Effective sealing of the mask, combined with respiratory filters should achieve 95% or greater filtration of particles less than 1um. Quantitative fit-testing (QNFT) data from these stop-gap devices has not been published to date. Our team developed one such device, the "SSM", and evaluated it using QNFT.

### Methods

Device prototypes were iteratively evaluated for comfort, breathability and communication, by team members wearing them for 15-30min. The fit and seal were assessed by positive and negative pressure user seal checks. The final design was then formally tested by QNFT, according to CSA standard Z94.4–18 in 40 volunteer healthcare providers. An overall fit-factor >100 is the passing threshold. Volunteers were also tested by QNFT on disposable N95 masks which had passed qualitative fit testing (QLFT) by institutional Occupational Health and Safety Department.

### Results

The SSM scored 3.5/5 and 4/5 for comfort and breathability. The median overall harmonic mean fit-factors of disposable N95 and SSM were 137.9 and 6316.7 respectively. SSM

**Funding:** The authors received no specific funding for this work.

**Competing interests:** The authors have declared that no competing interests exist.

scored significantly higher than disposable respirators in fit-test runs and overall fit-factors (p <0.0001). Overall passing rates in disposable and SSM respirators on QNFT were 65% and 100%. During dynamic runs, passing rates in disposable and SSM respirators were 68.1% and 99.4%; harmonic means were 73.7 and 1643.

## Conclusions

We present the design and validation of a reusable N95 stop-gap filtering facepiece respirator that can match existent commercial respirators. This sets a precedence for adoption of novel stop-gap N95 respirators in emergency situations.

## Introduction

The COVID-19 pandemic has unveiled widespread institutional shortages of N95 respirators and placed enormous strain on global supply chains. N95 respirators are critical pieces of personal protective equipment (PPE), and serve to mitigate the risk of air-borne exposure during aerosol-generating procedures. Access to adequate protection has been a key source of anxiety for healthcare workers in an unprecedented era of healthcare need and resource limitation [1, 2].

In response, there has been an international drive to develop novel stop-gap respirators that can be used when commercial N95 supplies are exhausted. One major challenge is the lack of standardized validation of non-commercial N95 replacements. Therefore, it has been difficult to prove whether non-commercially made reusable masks can potentially provide adequate filtration protection for frontline workers, in a COVID-19 pandemic surge scenario. One important aspect related to this is the relative performance of existent disposable N95 respirators compared to replacements. Recent literature suggests that there is a failure rate of over 25% in commercial disposable N95 respirators when tested quantitatively on actual workers [3, 4]. In particular, minimal information is available on respirator performance during talking, which is considered a particular challenge when fit-testing. Validation of any stop-gap solution necessarily implies comparison with the default standard level of protection.

Respirator design must also accommodate for the significant inter-individual variation in facial dimensions that exists within any workplace population. In 2019, the US National Institute for Occupational Safety and Health (NIOSH) adopted a Bivariate Panel as part of its evaluation of air-purifying respirators. The NIOSH Bivariate Panel is based on a large anthropometric survey of the US civilian population. It divides individuals into 10 different groups based on permutations of their facial length and width [5]. Respirators that are designed to fit individuals across the full spectrum of this panel, are anticipated to accommodate greater than 95% of the US workforce. Therefore, meeting the needs of a diverse and representative population is integral to the validation of any newly designed respirator.

This study sought to validate the filtration efficacy by quantitative fit-testing (QNFT) of a novel, reusable silicone-moulded face mask (SSM) that was developed by the University Health Network's Advanced Perioperative Imaging Lab (UHN APIL) as a potential stop-gap solution. Secondly, the performance of SSM was compared to the institutionally approved commercial disposable N95 model for each participant, in a cohort of healthcare workers. The disposable N95 models had passed qualitative fit testing (QLFT) by our institution's Occupational Health and Safety Department. Lastly, this study aims to establish a precedent methodology for non-commercial groups in evaluating future stop-gap respirator designs.

## Methods

We applied a phased approach to guide the design, refinement and validation of a novel, bio-safe silicone-moulded face mask (SSM). The phases represent a streamlined version of standardized methods to rapidly assess and iteratively improve prototype respirators, and include industry standard tests that must be performed prior to widespread application. The study was approved by the University Health Network Research Ethics Board, Toronto, under the study protocol ID 20–5435.0. All volunteers, including design and study team members, provided informed written consent for participation in the study. ("The corresponding author pictured in S1 Fig 'Striking Image' has provided written informed consent to publish his image alongside the manuscript").

### Design and production

Many designs employ 3D printing technology, silicone moulding or plastic extrusion to create mask bodies that can be decontaminated and reused. Dividing the mask design problem into two components namely seal and filtration, we developed prototypes addressing these two issues. When well-sealing silicone masks reverse engineered from patient face masks were paired with pleated-membrane respiratory filters, greater than 95% particulate filtration efficiency was achieved in prototype testing.

The SSM originated from the simple silicone respirator developed by Dr. Christian Petropolis at the University of Manitoba, and was adopted, redesigned and manufactured locally, using silicone injection moulding. Moulds were printed using PRUSA I3 MK3S (Prusa Research, Prague, Czech Republic). DragonSkin 20™ biosafe silicone (SmoothOn Inc., Macungie, PA) was used to form the mask body. This elastomeric mask provides seal and has a port for use with pleated-membrane respiratory filter, for which we elected the Intersurgical Air-Guard™ [6] (Burlington, Canada). 3D-printed harnesses in glycol-modified polyethylene terephthalate (PETG) or polyactic acid (PLA) plastic with attachments for elastic straps secured the masks onto wearers, see Fig 1. The design is under a CERN Open Hardware Licence Version 2—Strongly Reciprocal license and can be adopted and used by the public [7].

### Standardized subject validation

The keystone phase is the subject-validation by quantitative fit-testing using a standardized generated NaCl aerosol protocol. The efficacy of any newly developed respirator can be assessed with fit-testing and must meet relevant national regulatory standards. In Canada, this is set by the Canadian Standards Association (CSA) in Standard Z94.4–18 'Selection, Uses and Care of Respirators' [8]. This standard is equivalent to the Occupational Safety & Health Association (OSHA) regulation 1910.134 (f) (USA) [9]. Fit-testing may be categorized as either qualitative (QLFT) or quantitative (QNFT). Qualitative tests are based on the mask-wearer's ability to subjectively detect a challenge agent, such as denatonium benzoate, a bitter-tasting aerosolised solution (Bitrex™) [10]. QLFT is the standard subjective test performed at North American health centers including the authors' institution UHN. According to CSA Z94.4–18, each staff is to be fit-tested by QLFT before being assigned a respirator. In contrast, QNFTs utilize ambient particle counting devices to objectively measure the concentration of aerosols in the ambient environment versus the concentration inside the respirator.

Both QLFT and QNFT include seven "runs" or maneuvers, namely normal breathing, deep breathing, turning head side-to-side, nodding up-and-down, counting out loud, bending over, and ending with normal breathing again. In QLFT, a pass requires the subject to not detect the noxious stimulus in the seven runs. In QNFT, a pass requires the subject to score an overall fit-factor of 100 or greater, with at least six individual run score fit-factors of 100 or greater. In

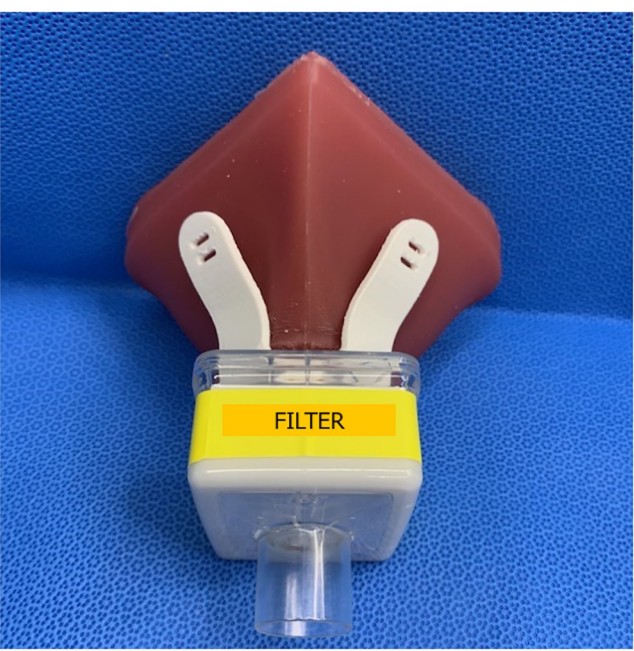

**Fig 1. SSM with harness (strapless) and Air-Guard™ filter.**

other words, a subject may score less than 100 in one of the runs, such as talking, and still pass overall.

The fit-factor is the ratio of aerosol particle concentration in the ambient air compared to the inside of the respirator, with integral count over time [11]. The overall fit-factor is defined by the OSHA as the harmonic mean of the seven individual run fit-factors [9].

$$Overall \ Fit \ Factor = \frac{N}{\frac{1}{ff1} + \frac{1}{ff2} + \frac{1}{ff3} + \cdots + \frac{1}{ffN}},$$

*where N = No. of Runs, which is 7 for QNFT per CSA Z94.4-18.*

According to Levitt-Safety documentation [12], the AccuFIT™ 9000 Quantitative Respirator Fit Tester complies with CSA Z94.4–18 and was therefore selected for QNFT subject validation. This tester detects aerosols, such as generated NaCl particles of size spectrum between 0.02 to 1.00 um, and thus quantifies the run fit-factors.

## Informal testing

The first step to assess the fit and function of a newly produced SSM respirator prototype was to don the mask and make informal assessments of comfort, fit and breathability. To do this, team members wore the mask for fifteen to thirty minutes during their regular, non-clinical work days. Wearers performed positive and negative pressure user seal checks according to CSA Z94.4–18, and repositioned or tightened the mask in the event of a failed seal check. During this trial period, note was made of the SSM's comfort, ease of breathing, speech clarity, head and neck mobility, and visibility. Next, we performed QNFT testing on two to three team members from the design and production team, to assess the quality of the SSM's fit. Shortcomings from any of these initial tests was used to inform and guide modifications by the design team, to refine further iterations of the mask design.

## Volunteer testing

To assess the SSM's fit across a wide range of individuals, we recruited 40 volunteers from the University Health Network (UHN) Toronto General Hospital's perioperative services. All participants were required to have passed standard Department of Occupational Health and Safety N95 QLFT within the last two years, and were asked to bring a clean N95 respirators for comparison. These respirator models were individually assigned to the healthcare worker after mandatory institutional QLFT testing using Bitrex™ challenge, with fit validation for two years according to institutional and CSA Z94.4–18 protocol. The exclusion criteria were those who did not have a valid QLFT fit-test assigned N95 respirator, flu-like symptoms, and those who were unshaven or unable to don on/off an N95 or a silicone face mask for other reasons.

After informed consent, demographic data was recorded, including age, gender, height, weight, and morphometric measurements were taken, including menton-sellion length, face-height and face-width, the latter two used in the NIOSH face bivariate panel [5]. Face-height is the distance between the bridge of the nose and the bottom of chin; face-width is the maximal bizygomatic distance; menton-sellion length is the distance between the prominence of nasal bridge to the prominence of the chin [13].

Using an AccuFIT 9000™ (Levitt-Safety, Oakville, Canada), two testers (WN, AS) performed sequential QNFT testing according to CSA Z94.4–18 on the volunteer's 3M N95 and then the SSM. Masks were probed as per industry standard, and users were instructed to perform positive and negative pressure user seal checks. Any physical adjustment to the fit of either mask was carried out at this time. QNFT testing was performed following the AccuFIT 9000's built-in testing environment. Fit-factor measurements were recorded for each of the seven standard "runs" or maneuvers, according to the CSA Z94.4–18 standard listed above. For the purposes of this study, grimace testing was not performed, as it is not required according to this standard for half-face filtering respirators. Maneuvers were categorized as either stationary (normal breathing, deep breathing) or dynamic (turning, nodding, talking, bending) for further analysis. The overall fit-factor–the harmonic average of the seven individual measurements–was recorded, and a score greater than 100 was deemed acceptable, as required by the CSA standard for tight-fitting half-face filtering respirators. All volunteers were also asked to rate the SSM's comfort and breathability on a five-point scale.

## Statistical analysis

Statistical analysis was performed using Stata 14 (StataCorp. 2015, College Station, TX), and RStudio version 1.3 (RStudio 2020, Boston, MA) for data visualizations. Univariate analysis was performed on demographic and risk factors influencing commercial mask fit failure. The non-parametric Wilcoxon signed-rank test was used to compare differences in fit-factor scores of paired runs of equal sample size; the non-parametric Kruskal–Wallis test was used to compare differences in fit-factor scores of different sample sizes. Median overall fit-factors were chosen as summary statistics to compare the mask performances in the group. The sample size of 40 was selected firstly by estimation of the failure rate of less than one percent of the SSM and 33 percent of the commercial N95, according to known studies [3, 4] of N95 by QLFT and internal reviews assessing N95 performance. Sample size calculation assuming two groups, alpha of 0.05 and power of 0.8, suggested a minimum of 40 volunteers to detect difference in overall pass rates. When each "run" of the QNFT was treated as an observation, then 40 fit-tests results in 280 paired runs, and this provides sufficient power to detect difference in pass rates of runs.

## Results

### Cohort characteristics

The cohort of 40 participants had a mean age of 38.7 (9.87), and there were equal numbers of male and female participants (Table 1). The mean BMI was 23.39 (2.71), with 70% of participants characterized as underweight/normal weight based on BMI categorization. A wide-range of anthropometric facial dimensions was exhibited, with 9 of 10 NIOSH panels represented amongst the cohort. Five participants (12.5%) demonstrated measurements outside of the standard NIOSH panel range. The volunteer healthcare workers also displayed variation in their assigned (previously QLFT fitted) disposable N95 respirators, using four different models of 3M™ N95 respirators, namely 1860 (25%), 1860S (17.5%), 1870+ (47.5%), and 8210 (10%) (Table 1).

### Baseline characteristics on 3M N95 respirator fit

Similar to the literature, 65% (26/40) of the participants achieved an overall "pass" with their previously-fitted disposable N95 respirators. Table 2 represents the distribution of participant

**Table 1. Demographics, anthropometric characteristics, and qualitatively fitted 3M N95 model.**

| Participant Demographics | N = 40 |
|---|---|
| Age, mean (SD), y | 38.7 (9.87) |
| Female–no./total (%) | 20/40 (50%) |
| Body Mass Index, categorical no./total (%) | Under-weight 1/40 (2.5%) † |
| | Normal weight 27/40 (67.5%) |
| | Overweight 8/40 (20%) |
| | Obese 4/40 (10%) |
| Body Mass Index, mean (SD), kg/m$^2$ | 23.39 (3.71) |
| **Anthropometric** | |
| Face Width mean (SD), mm | 132.35 (9.99) |
| Face Length mean (SD), mm | 119.52 (8.3) |
| Menton-sellion distances mean (SD), mm | 98.1 (16.74) |
| **NIOSH panel** | |
| 1 no./total (%) | 2/40 (5%) |
| 2 no./total (%) | 2/40 (5%) |
| 3 no./total (%) | 7/40 (17.5%) |
| 4 no./total (%) | 3/40 (7.5%) |
| 5 no./total (%) | 2/40 (5%) |
| 6 no./total (%) | 6/40 (15%) |
| 7 no./total (%) | 8/40 (20%) |
| 8 no./total (%) | 0/40 (0%) |
| 9 no./total (%) | 3/40 (7.5%) |
| 10 no./total (%) | 2/40 (5%) |
| NA no./total (%) | 5/40 (12.5%) |
| **3M N95 Model** | |
| 1860 no./total (%) | 10/40 (25%) |
| 1860S no./total (%) | 7/40 (17.5%) |
| 1870+ no./total (%) | 19/40 (47.5%) |
| 8210 no./total (%) | 4/40 (10%) |

† Only one participant had a BMI is less than 18.5 and fell within the underweight range. For statistical models, the underweight BMI and normal weight categories were lumped together to give a single category.

**Table 2. Demographic anthropometric characteristics and 3M N95 model distribution based on success of 3M N95 fit test.**

| | 3M N95 Test | |
|---|---|---|
| | **Pass no./total (%)** | **Fail no./total (%)** |
| | **26/40 (65%)** | **14/40 (35%)** |
| **Demographic** | | |
| Age mean (SD), y | 37.92 (8.55) | 40.14 (12.17) |
| Female sex–no./total (%) | 13/26 (50%) | 7/14 (50%) |
| BMI, mean (SD), kg/m$^2$ | 23.59 (3.79) | 23.04 (3.66) |
| **BMI** | | |
| Under/normal weight no./total (%) | 19/26 (73,08%) | 9/14 (64,28%) |
| Overweight no./total (%) | 4/26 (15,38%) | 4/14 (28,57%) |
| Obese no./total (%) | 3/26 (11,54%) | 1/14 (7,14%) |
| **Anthropometric** | | |
| Face width mean (SD), mm | 131.69 (8.37) | 133.57 (12.73) |
| Face length mean (SD), mm | 119.35 (6.87) | 119.86 (10.77) |
| Menton-sellion distance mean (SD), mm | 99.56 (7.65) | 95.36 (26.75) |
| **NIOSH panel** | | |
| 1 no./total (%) | 1/24 (4.17%) | 1/11 (9.09%) |
| 2 no./total (%) | 1/24 (4.17%) | 1/11 (9.09%) |
| 3 no./total (%) | 5/24 (20.83%) | 2/11 (18.18%) |
| 4 no./total (%) | 2/24 (8.33%) | 1/11 (9.09%) |
| 5 no./total (%) | 1/24 (4.17%) | 1/11 (9.09%) |
| 6 no./total (%) | 6/24 (25%) | 0/11 (0%) |
| 7 no./total (%) | 7/24 (29.16%) | 1/11 (9.09%) |
| 9 no./total (%) | 1/24 (4.17%) | 2/11 (18.18%) |
| 10 no./total (%) | 0/24 (0%) | 2/11 (18.18%) |
| **3M N95 model** | | |
| 1860 no./total (%) | 4/26 (15.38%) | 6/14 (42.86%) |
| 1860S no./total (%) | 4/26 (15.38%) | 3/14 (21.43%) |
| 1870+ no./total (%) | 14/26 (53.85%) | 5/14 (35.71%) |
| 8210 no./total (%) | 4/26 (15.38%) | 0/14 (0%) |

demographics, anthropometric characteristics, and types of 3M N95 respirators. The passing rate during the talking maneuver was 57.5% (23/40). The passing rate for dynamic maneuvers was 68.1% (109/160), and for stationary maneuvers 74.2% (89/120). Univariate logistic regression did not reveal any statistically significant association between the 'likelihood of passing the fit test' and baseline characteristics such as age, BMI category, gender, facial dimensions, NIOSH panel, or 3M N95 mask model (Table 3).

## Quantitative performance of SSM vs. 3M N95 models

The SSM showed an overall success rate of 100%, with all 40 volunteers achieving an acceptable fit. In comparison, 65% of the participants achieved an overall "pass" with their assigned disposable N95 respirators. Heat maps illustrating individual participant success rates across the seven different maneuvers are depicted in Figs 2 and 3, representing disposable N95 and SSM tests for the 40 volunteers. Whereas 17/40 participants scored fit-factors less then 100 during talking with disposable N95 respirators, only 1 of 40 participants scored less than 100 (actual score 90.9) with the SSM.

**Table 3. Univariate binary and ordinal logistic regression analyses for association of 3M N95 success rate and baseline characteristics of volunteers.**

| | Logistic Regression | | |
|---|---|---|---|
| | OR | 95%CI | P-value |
| **Demographics** | | | |
| Age | 0.98 | [0.92–1.04] | 0.5 |
| Gender (F = 0, M = 1) | 1 | [0.27–3.67] | 1 |
| BMI | | | |
| Under/normal weight | Reference | Reference | Reference |
| Overweight | 0.47 | [0.96–2.34] | 0.36 |
| Obese | 1.42 | [0.13–15.63] | 0.77 |
| **Anthropometrics** | | | |
| Face width | 0.98 | [0.92–1.05] | 0.57 |
| Face length | 0.99 | [0.92–1.07] | 0.85 |
| Menton-sellion distance | 1.01 | [0.97–1.06] | 0.46 |
| **NIOSH Panel †** | | | |
| 1 | Reference | Reference | Reference |
| 2 | 1 | [0.2–50.4] | 1 |
| 3 | 2.5 | [0.1–62.6] | 0.58 |
| 4 | 2 | [0.51–78.25] | 0.71 |
| 5 | 1 | [0.2–50.4] | 1 |
| 7 | 7 | [0.22–226] | 0.27 |
| 9 | 0.5 | [0.12–19.56] | 0.71 |
| **3M Models †** | | | |
| 1860 | Reference | Reference | Reference |
| 1860S | 2 | [0.28–14.2] | 0.49 |
| 1870+ | *4.2* | *[0.83–21.35]* | *0.084* |

† NIOSH Panels 6 and 10 as well as 3M model 8210 predicts success or failure perfectly and was omitted from the logistic regression.

The median overall fit-factors were 6317 and 137.9 for the novel SSM and disposable respirators respectively. This represents an approximately $1.6^*\log_{10}$ times difference in median fit-factors (Fig 4). Visualization of pairwise comparison of $\log_{10}$ adjusted overall fit-factors are shown in Figs 5 and 6. Overall fit-factors were significantly higher in the SSM, using a Wilcoxon matched-pairs signed-rank test (p < 0.0001, Table 4). This finding was reflected across the fit-factor measurements for all seven separate maneuvers–normal breathing, deep

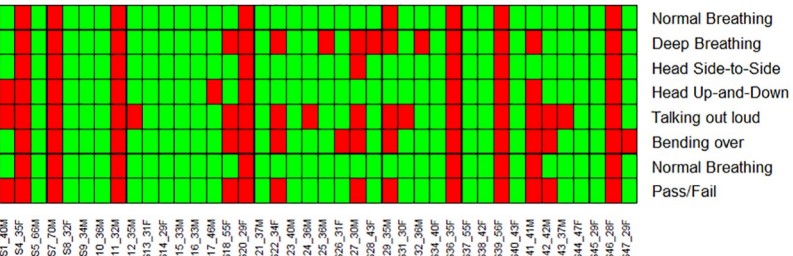

**Fig 2. Representation of success rate of 3M N95 throughout the 7 runs of test on 40 participants.** Green indicates pass (fit-factor of 100 or greater) and red indicates fail (fit-factor less than 100).

**APIL SSM prototype**

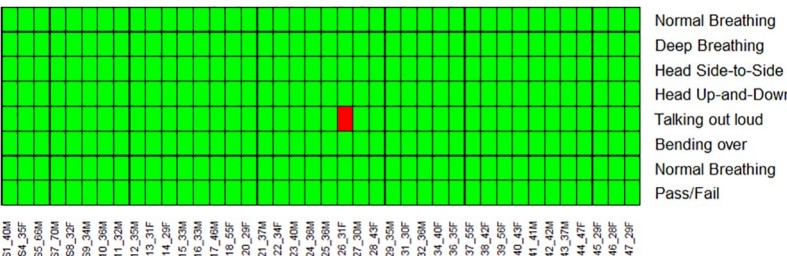

**Fig 3. Representation of success rate of APIL SSM throughout the 7 runs of test on 40 participants.** Green indicates pass (fit-factor of 100 or greater) and red indicates fail (fit-factor less than 100).

breathing, turning, nodding, talking, bending, and normal breathing repeated, with p values < 0.0001 for all run comparisons (Table 5). The novel SSM scored significantly higher in a composite (harmonic mean) of the three static maneuvers (3832 vs. 89.2, p < 0.0001) as well as the four dynamic maneuvers (1643 vs. 74.7, p < 0.0001) when compared to the disposable respirators (Table 6). Within each respirator group, SSM scored significantly higher in static maneuvers compared to dynamic maneuvers (Kruskal-Wallis equality of populations rank test, p = 0.0001) but this difference was attenuated and non-significant in disposable respirators (p = 0.1168, Table 6).

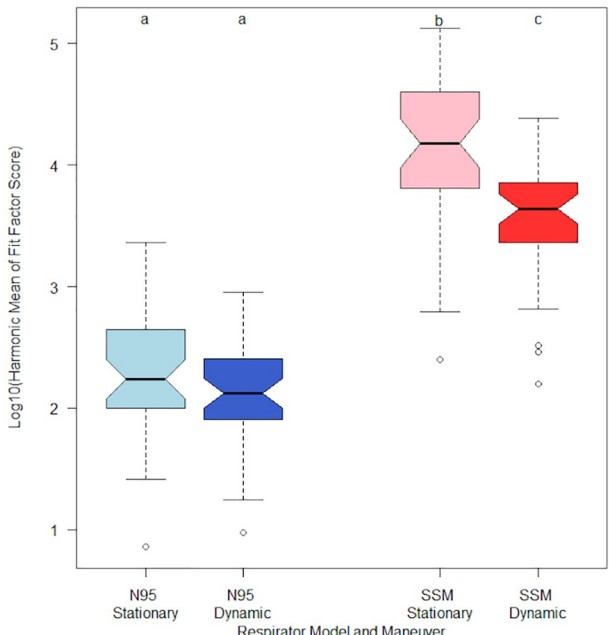

**Fig 4. Boxplot of 3M N95 and SSM stationary and dynamic Log$_{10}$(harmonic mean) fit-factors. ‡.** ‡ The lower and upper box boundaries are the first (Q1) and third (Q3) quartiles, respectively, band near the middle of the box is the median (second quartile). The upper whiskers are located at the smaller of the maximum value and Q3 + 1.5 interquartile range (IQR), whereas the lower whiskers are located at the larger of the smallest value and Q1–1.5 IQR. Empty circles are outliers that fall outside this range.

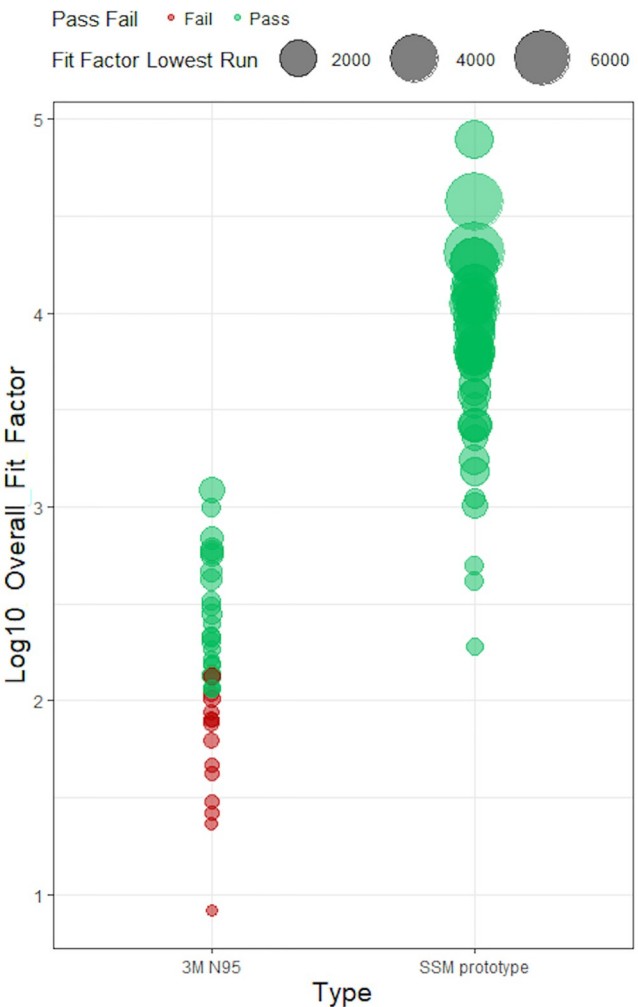

**Fig 5. Log₁₀ group comparison between 3M N95 and APIL SSM overall fit factors for 40 participants.** The overall fit-factor was defined as the harmonic mean of the seven individual run fit-factors.

## Participant appraisal of SSM

Qualitative appraisal ratings of the novel SSM demonstrated an average score of 3.5 for comfort (range 2 to 5) and an average score of 4 for breathability (range 2 to 5).

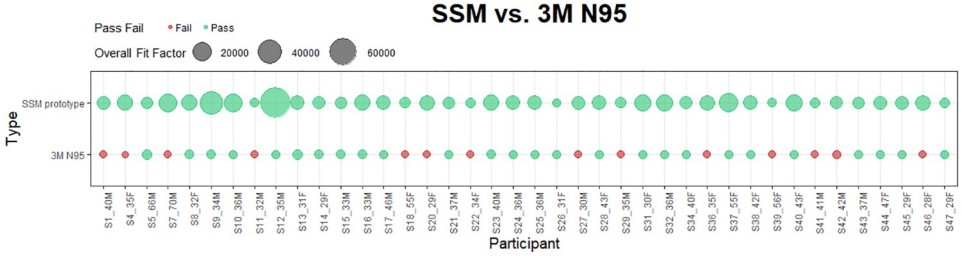

**Fig 6. Pairwise comparison of 3M N95 and APIL SSM overall fit-factors for 40 participants.**

**Table 4. Comparison of harmonic means of the individual maneuver fit-factors between 3M N95 and APIL SSM using Wilcoxon signed-rank test.**

| | 3M N95 Harmonic Mean [95% conf. interval] | APIL SSM Harmonic Mean [95% conf. interval] | P value (Wilcoxon Signed-Rank test) |
|---|---|---|---|
| **Normal Breathing** | 100.8 [61.3–282.0] | 3066.0 [undefined CI]‡ | P < 0.0001 |
| **Deep Breathing** | 76.6 [44.0–295.8] | 3372.2 [2007.0–10546.2] | P < 0.0001 |
| **Head Side-to-side** | 93.5 [57.7–245.5] | 2026.2 [undefined CI] ‡ | P < 0.0001 |
| **Head Up-and-Down** | 87.6 [58.6–173.1] | 1679.3 [1001.0–5208.2] | P < 0.0001 |
| **Talking out loud** | 77.8 [60.0–110.6] | 932.5 [581.6–2350.6] | P < 0.0001 |
| **Bending over** | 53.9 [34.7–120.4] | 3658.8 [2143.4–12485.3] | P < 0.0001 |
| **Normal Breathing** | 93.9 [60.1–213.8] | 6246.3 [3452.8–32707.5] | P < 0.0001 |

‡ Due to some extremely high fit-factors close to infinity or 99999999.

**Table 5. Comparison between 3M N95 and APIL SSM using Wilcoxon signed-rank test.**

| Maneuvers † | | Neg/Pos/Ties‡ | Observer N˚ | z | p |
|---|---|---|---|---|---|
| **N95 all runs (reference) vs SSM all run (comparison)** | Neg | 2 | 14.49 | < 0.0001 |
| | Pos | 278 | | |
| | Ties | 0 | | |
| **N95 stationary runs (reference) vs SSM stationary runs (comparison)** | Neg | 0 | 9.51 | < 0.0001 |
| | Pos | 120 | | |
| | Ties | 0 | | |
| **N95 dynamic runs (reference) vs SSM dynamic runs (comparison)** | Neg | 2 | 10.95 | < 0.0001 |
| | Pos | 158 | | |
| | Ties | 0 | | |
| **N95 stationary harmonic average (reference) vs SSM stationary harmonic average (comparison** | Neg | 0 | 5.51 | < 0.0001 |
| | Pos | 40 | | |
| | Ties | 0 | | |
| **N95 dynamic harmonic average (reference) vs SSM dynamic harmonic average (comparison)** | Neg | 0 | 5.51 | < 0.0001 |
| | Pos | 40 | | |
| | Ties | 0 | | |
| **N95 overall fit factor (reference) vs SSM overall fit factor (comparison)** | Neg | 0 | 5.51 | < 0.0001 |
| | Pos | 40 | | |
| | Ties | 0 | | |

† All runs (7 maneuvers), stationary runs (3 maneuvers: normal breathing, deep breathing, normal breathing repeated), dynamic runs (4 maneuvers: head side-to-side, head up-and-down, counting out loud, bending over).

‡ Negative means SSM scored less than disposable N95, positive means SSM scored higher than disposable N95.

**Table 6. Comparison between stationary harmonic runs and dynamic harmonic runs of N95 and APIL SSM using Wilcoxon signed rank test (between groups) and Kruskal Wallis rank test (within groups).**

| | 3M N95 | APIL SSM | P-value (Wilcoxon Signed Rank test) |
|---|---|---|---|
| **Stationary Runs Harmonic Mean [95% conf. interval]** | 89.2 [54.4–247.4] | 3832.4 [2110.3–20831.2] | P < 0.0001 |
| **Dynamic Runs Harmonic Mean [95% conf. interval]** | 74.7 [51.5–136.2] | 1642.9 [1018.1–4253.5] | P < 0.0001 |
| **P-value (Kruskal Wallis Rank Test)** | P = 0.1168 | P < 0.0001 | |

## Discussion

One of the original project goals of the team was to assess the feasibility of developing a reusable N95 grade mask, using computer aided design and printing techniques such as 3D printing with PETG or PLA, off-the-shelf components such as biosafe silicone, and medical grade respiratory filters. Given the highly significant overall fit-factor and individual run fit-factor comparison, we believe that stop-gap solutions are feasible. Our device can be locally manufactured at a cost of CDN $15–20 of materials and 20 minutes of labour (assuming a run of 1000). Further usability testing to evaluate work of breathing and voice conduction are required to fully characterise and capabilities of this device.

Initial and potential road blocks for community groups include design know-how, experience with use of 3D printing, and silicone pouring and other manufacturing techniques. At the time of writing, our authoring group has produced online access to working files which has been taken-up by overseas medical device groups for testing; a paper detailing the production process of the SSM is also under review. Access to medical-grade respiratory filters may be difficult in a crisis scenario, especially if such filters are also utilized in filtration for ventilators. We stress the need to conserve PPE and respiratory device supplies, and increased reusability of both mask body and filter is one such solution in a prolonged pandemic course.

Within eight weeks using a rigorous iterative process, we were able to develop an open source solution that meets formal requirements for filtration efficiency and filtration area using a pleated-membrane medical respiratory filter. The effective surface area of the Intersurgical Air-Guard™ pleated-membrane filter is on the order of 1050 sq cm (150 cm x 7 cm). This is significantly greater than the filtration surface area of all disposable models that we are aware of. This is the first reported human subject validation of a locally manufactured reusable respirator using standardized quantitative fit testing (per CSA Z94.4–18). Within the same collaborative, a 2nd N95 respirator derivative has been developed that contains valves that prolong filter life, decrease subjective work of breathing, and improve acoustics [14]. Industry partners affiliated with the collaborative have followed suit and developed industry grade masks that can be similarly validated. The baseline performance (less than 70% overall pass rate) of commercial disposable N95 respirators is consistent with recent literature [3, 10]. QLFT is subjective, dependent on concentration of stimulant aerosol under-the-hood, and has a significant false negative rate [4]. QNFT gives the wearer numerical feedback, an opportunity to know under which dynamic conditions they are at risk, and an incentive to fit masks properly. After fit-test failures of disposable respirators, our testers would spend extra time beyond protocol guidelines in trying to improve fitting by education, refit onto different mask models, in order to improve overall performance, and, as a last resort, refer to Occupational Health and Safety for non-disposable alternatives. A future quality control study question is: "Of those who fail their assigned mask, how many can be retrained to re-pass a 2nd or 3rd QNFT on the same respirator and/or another model?" UHN APIL and The Department of Anesthesiology, are in the process of preparing for such a quality-improvement audit, of utmost importance in the 2nd COVID-19 pandemic wave across Canada [15]. Of particular concern is whether our volunteer facial sizing is comparable to the community, given the average BMI was approximately 24 kg/m$^2$. There were five volunteers whose face belonged to the NIOSH panel 9 and 10, the widest and longest of the facial type panel: all five passed on the SSM. Given that the SSM relies on a silicone rim seal, where there is more skin surface for rim contact there will be improved respirator seal. As a result, seal quality would be expected to positively correlate with BMI. Once again, this emphasizes the need for QNFT on the individual, even when given highly efficacious respirators [16].

A related concern is the disparity between the apparent performance of gold standard commercial disposable N95 respirators and the SSM fitted with a pleated-membrane respiratory filter. We have already mentioned the great disparity in gas exchange surface area, with the Intersurgical Air-Guard™ filter having much greater exchange area compared to that of the disposable N95 respirators available to our volunteers. Once again, mask seal proves to be of utmost importance in both testing and clinical situations, with silicone offering potentially better seals compared to paper masks.

Further evaluation of the mask will include testing for compliance with NIOSH standards (42 CFR, Part 84, Subpart K, §84.181) [17]. This test evaluates performance of the mask under standard conditions on a mannequin with a sampling port. That means 84 L/min flow for single filter cartridge, with option to split up to a maximum of three cartridges, using aerosolized NaCl $0.075 \pm 0.020$ micrometer at density not exceeding 200 mg/m$^2$, at chamber temperature $38 \pm 2.5°$C and $85 \pm 5\%$ relative humidity; see reference for full details on testing conditions. The respirator filtration efficiency of aerosolized NaCl at 0.3 um diameter should be 95% or greater, with 0.3um being the largest penetrating [18]. This is a costly commercial test and we are currently seeking a certified laboratory and funding to complete this process. However our team and institutional occupational health experts are confident that given the documented performance of the Intersurgical Air-Guard™ filter (a 99.999% filtration rate with minimal pressure drop) [6] the device is highly likely to pass this testing. The AccuFIT system contains a particle counter capable of detecting particle sizes within the 0.02 to 1 um range. For droplet sizes beyond this, it is assumed that if the system is efficient in this particle range then it is also efficient for ranges from 1 to 5 um, a common droplet size produced by human expectoration [4]. This is a reasonable assumption given the particular pleated-membrane respiratory filter we appended to the SSM. We advise that stop-gap solutions refer to the product information of the respiratory filter attached, and if there is doubt, request commercial testing to extend the particle test ranges beyond the 1 um limit.

Health regulations at the state, provincial, and/or national level oversee PPE and respirator deployment to ensure frontline worker safety and public health [19]. For any non-commercially certified PPE including respirators that have not the resources to undergo speedy NIOSH and other testing, Health Canada has provided guidelines for an accelerated temporary approval process [20].

The SSM used in this study was of one (medium) size and was able to provide an effective seal on a wide range of faces. However, it is anticipated that mask comfort would be improved by more tailored sizing. Feedback from participants included requests for a smaller size, lighter total weight, less heat and moisture build-up, and clearer audio quality. Refinement of device usability and reliable donning and doffing techniques requires involvement of human factors specialists. In addition to effective filtration and usability, the end design must allow reliable, rapid production at modest costs using accessible materials with small-scale distributed manufacturing processes and other low entry-barrier techniques, in order to be accessible to the majority of institutions and health care providers who may benefit. Rigorous and user-friendly training protocols, developed by specialists in human factors design and evaluation, are required for effective, institution wide protection of health care workers. Respiratory protection relies ultimately on competently trained wearers [21].

## Conclusion

Production of a reusable N95-equivalent stop-gap respirator that can be manufactured locally is not only feasible, but has the potential for exceeding commercial disposable respirator performance. The use of quantitative fit-testing on individual wearers is important in subject

validation and deployment of any novel respirator. It can educate the wearer on specific situations for field-use failure. Further clinical research of novel reusable N95 solutions should inform specific points for improvement that maximize user performance and comfort. Research on prolonged filter use, filter preservation, and potential for filter decontamination and reuse will strengthen similar stop-gap respirator designs. This is a timely endeavour that the COVID-19 pandemic has both encouraged and necessitated.

## Supporting information

**S1 Fig.**
(TIF)

**S1 Dataset.**
(XLSX)

## Acknowledgments

We thank the Department of Anesthesiology and Pain Management, and the Joint Department of Medical Imaging, at University Health Network for their academic support. In addition, we would like to thank Dr. Thomas Looi, Director of SickKids CIGITI, for crucial advice on design reiterations; Dr. David Green, from UHN's Techna Institute, for lending space for fit-testing; Intersurgical (Burlington, Canada) for their donation of Air-Guard™ respiratory filters. Special thanks to Nasa Chau Nguyen (BASc) and Natasha Valenton (BASc) from The School of Applied Science and Engineering, Andy Efemu (BSc) from The MD Program, University of Toronto, for his invaluable work on the designs and revisions of the SSM and its derivatives.

## Author Contributions

**Conceptualization:** William C. K. Ng, Vahid Anwari, Azad Mashari.

**Data curation:** William C. K. Ng, Andrew Syrett, Zixuan Xiao.

**Formal analysis:** William C. K. Ng, Arnaud Romeo Mbadjeu Hondjeu, Zixuan Xiao.

**Funding acquisition:** William C. K. Ng.

**Investigation:** William C. K. Ng, Andrew Syrett, Dmitry Rozenberg, Vahid Anwari.

**Methodology:** William C. K. Ng.

**Project administration:** William C. K. Ng, Vahid Anwari.

**Resources:** William C. K. Ng, Vahid Anwari, Azad Mashari.

**Software:** Zixuan Xiao.

**Supervision:** William C. K. Ng, Azad Mashari.

**Validation:** William C. K. Ng.

**Visualization:** William C. K. Ng, Arnaud Romeo Mbadjeu Hondjeu, Zixuan Xiao.

**Writing – original draft:** William C. K. Ng, Arnaud Romeo Mbadjeu Hondjeu, Andrew Syrett, Rebecca Caragata.

**Writing – review & editing:** William C. K. Ng, Arnaud Romeo Mbadjeu Hondjeu, Rebecca Caragata, Dmitry Rozenberg, Zixuan Xiao, Vahid Anwari, Jessica Trac.

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
