## [Decision Letter · Decision Letter 0]

5 Oct 2020

PONE-D-20-18656

Subject validation of reusable N95 stop-gap filtering face-piece respirators in COVID-19 pandemic

PLOS ONE

Dear Dr. Ng,

Thank you for submitting your manuscript to PLOS ONE. After careful consideration, we feel that it has merit but does not fully meet PLOS ONE’s publication criteria as it currently stands. Therefore, we invite you to submit a revised version of the manuscript that addresses the points raised during the review process.

We look forward to receiving your revised manuscript.

Kind regards,

Vahid Serpooshan, PhD

Academic Editor

PLOS ONE

Journal Requirements:

3. Please clarify in your Data availability statement how other researchers can access the dataset and open hardware used in this study. Please note that making data available on request from a single author is not an acceptable form of data sharing, according to PLOS ONE data availability policy https://journals.plos.org/plosone/s/data-availability.

4. We note that Figures in your submission contain copyrighted images.

All PLOS content is published under the Creative Commons Attribution License (CC BY 4.0), which means that the manuscript, images, and Supporting Information files will be freely available online, and any third party is permitted to access, download, copy, distribute, and use these materials in any way, even commercially, with proper attribution. For more information, see our copyright guidelines: http://journals.plos.org/plosone/s/licenses-and-copyright.

a. You may seek permission from the original copyright holder of Figures to publish the content specifically under the CC BY 4.0 license.

5. We note that the Striking Image Figure includes an image of a participant.

As per the PLOS ONE policy (http://journals.plos.org/plosone/s/submission-guidelines#loc-human-subjects-research) on papers that include identifying, or potentially identifying, information, the individual(s) or parent(s)/guardian(s) must be informed of the terms of the PLOS open-access (CC-BY) license and provide specific permission for publication of these details under the terms of this license.

Please download the Consent Form for Publication in a PLOS Journal (http://journals.plos.org/plosone/s/file?id=8ce6/plos-consent-form-english.pdf). The signed consent form should not be submitted with the manuscript, but should be securely filed in the individual's case notes.

Please amend the methods section and ethics statement of the manuscript to explicitly state that the patient/participant has provided consent for publication: “The individual in this manuscript has given written informed consent (as outlined in PLOS consent form) to publish these case details”.

6. Please clarify in your Ethics statement and Methods section the full name of the ethics committee that approved the study, along with the approval number.

Please also clarify whether the ethics approval covered both the volunteers and the team members.

Please also clarify how both volunteers and team members gave consent.

Please also clarify what is meant by 'team members'.

7. Please ensure that you refer to Figures 2 and 3 in your text as, if accepted, production will need this reference to link the reader to each figure.

Reviewers' comments:

Reviewer's Responses to Questions

**Comments to the Author**

1. Is the manuscript technically sound, and do the data support the conclusions?

Reviewer #1: Partly

Reviewer #2: Yes

2. Has the statistical analysis been performed appropriately and rigorously? 

Reviewer #1: Yes

Reviewer #2: Yes

3. Have the authors made all data underlying the findings in their manuscript fully available?

Reviewer #1: Yes

Reviewer #2: Yes

4. Is the manuscript presented in an intelligible fashion and written in standard English?

Reviewer #1: Yes

Reviewer #2: Yes

5. Review Comments to the Author

Reviewer #1: • A lot of COVID-19 containing droplets have been reported to be in 1-5um as well. The described tests do not address this potential infection range, so the authors should either comment on whether the tests that they chose would by default be successful in stopping the larger droplets as well, or perform a round of tests looking these larger droplets.

• Can the authors comment further on whether overweight/obese users would be able to benefit from the silicone mask in similar efficacy, since that is also a large demographic of potential users and the cohort used in this study skewed significantly towards the more uncommon underweight/normal range.

• The numerical values between the N95 and the SSM are vastly different. I felt that there needs to be further specific elaboration on why the values are so disparate and clearer explanation on what let to these results.

• While a valid alternative to commercial N95s, the authors need to be careful when claiming this is a straightforward or simple N95 alternative to produce. To manufacture the SSM, it requires a relatively well outfitted manufacturing facility and know-how that is not always common outside of large cities/institutions (silicon moulding, injection moulding, 3D design, etc) and the supplies that were used are also scarce or unavailable to the general public. There needs to be further elaboration on how this potential roadblock to production at scale can be overcome. Either by simplifying the design and using alternative materials, or by directly enlisting industry partners that have the capacity and ability to produce the SSM.

Reviewer #2: The paper presents on the validation approach for reusable respirators that can be used at hospitals and front line workers to access the reliability, reusability of N95 masks with replaceable filters. The paper is timely and should be published as is.

6. PLOS authors have the option to publish the peer review history of their article (what does this mean?). If published, this will include your full peer review and any attached files.

Reviewer #1: No

Reviewer #2: No

---

## [Author Response · Author response to Decision Letter 0]

28 Oct 2020

Dr. Vahid Serpooshan, PhD

Academic Editor

PLOS ONE

Toronto, Oct 12th, 2020

Re: Response to Reviewers of “Subject validation of reusable N95 stop-gap filtering facepiece respirators in COVID-19 pandemic”

ID: PONE-D-20-18656

Dear Dr. Vahid Serpooshan, Reviewer 1, and Reviewer 2,

Thank you all for all your time in reviewing this submission. The editorial and review comments are invaluable and are addressed sequentially, as outlined in the decision letter 5 Oct 2020:

1. We have reviewed the PLOS ONE style and formatting guidelines and have revised the Authors’ List, manuscript (MS), and Supporting Info Files as requested.

2. We have thoroughly copyedited the MS for language, spelling, grammar.

a. William Ng, Jessica Trac, Azad Mashari were involved in the rewriting of the MS,

b. A copy of MS with tracked changes has been uploaded as a supporting info file,

c. A clean copy of MS has been uploaded as the new *manuscript* file. All authors have reviewed the manuscript and resubmission.

3. Our Data availability statement has been clarified and as there are no restrictions, the minimal anonymized data set necessary to replicate the results has been added as a Support Info File and also made available on a public repository on our project github:

https://github.com/tgh-apil/Reusable-N95-Respirator/tree/master/10-Publication

4. The figure (Figure 1) that supposedly contains copyright image is not a copyrighted image, but an original one taken by the authors with a phone camera. We can, in addition, redact the image such that the product label is blurred, available as the uploaded file “Fig1 redacted.TIF”.

5. The striking image is a side profile of the corresponding author to demonstrate the fit of the stop-gap respirator. I have downloaded the Consent Form and filed in my case notes. The methods and ethics statement of the MS has made it explicit that I have provided consent for its publication.

6. The University Health Network Research Ethics Board approved the study protocol 20-5435.0, and 20-5435.0 is the assigned approval number. 

a. Team members were the members of the design and testing team, which has been clarified in the MS,

b. Team members and volunteers gave written consent,

c. Ethics approval covered both the volunteers and team members.

7. Figures 2 and 3 are now referenced clearly in the MS.

Response to Reviewer 1’s comments:

Thank you for the detailed comments, they were instructive and pointed out areas of improvement to the MS:

1. “• A lot of COVID-19 containing droplets have been reported to be in 1-5um as well. The described tests do not address this potential infection range, so the authors should either comment on whether the tests that they chose would by default be successful in stopping the larger droplets as well, or perform a round of tests looking these larger droplets.”

Thank you for pointing out the 1-5um droplet range and its potential for transmission of infection. The 1um was the upper limit of the AccuFIT detection. Larger droplets are at a less penetrating particle width, with the most penetrating particle width at 0.3um. So, we are in agreement with Abbott Safety and the standard of the AccuFIT tester in their choice of testing particle size range. Filtration at the most penetrating size does imply filtration of larger droplet sizes. We have added this under methods and also in the discussion.

2. “• Can the authors comment further on whether overweight/obese users would be able to benefit from the silicone mask in similar efficacy, since that is also a large demographic of potential users and the cohort used in this study skewed significantly towards the more uncommon underweight/normal range.”

The fitting of obese mask wearers is important. Our sample of participants captured a less obese population compared to standard North American BMI values, but it is consistent with healthcare workers. The comfort we take is that the SSM is an elastomeric mask, and more obese wearers is associated with more facial tissue, which is helpful to better respirator rim seal. We also note that the NIOSH panel subtypes 9 and 10 represent the longest and widest face types. This reiterates one of our main recommendations of the importance of quantitative fit testing (QNFT) on the individual wearer, especially with suspected fit-failure, and despite having high-quality respirators.

3. “• The numerical values between the N95 and the SSM are vastly different. I felt that there needs to be further specific elaboration on why the values are so disparate and clearer explanation on what let to these results.”

Yes, the QNFT results of the SSM is superior to the commercial disposable N95 respirators in testing conditions. The main contributors were the superior respirator seal (silicone vs paper rim) and greater surface area for gas exchange (already mentioned in MS discussion). We have expanded these two points in the discussion, and hope that will suffice. We also did not want to denigrate existent disposable respirator options, but merely wish to highlight that it is feasible to have comparable (sometimes better) reusable respirator alternatives.

4. “• While a valid alternative to commercial N95s, the authors need to be careful when claiming this is a straightforward or simple N95 alternative to produce. To manufacture the SSM, it requires a relatively well outfitted manufacturing facility and know-how that is not always common outside of large cities/institutions (silicon moulding, injection moulding, 3D design, etc) and the supplies that were used are also scarce or unavailable to the general public. There needs to be further elaboration on how this potential roadblock to production at scale can be overcome. Either by simplifying the design and using alternative materials, or by directly enlisting industry partners that have the capacity and ability to produce the SSM.”

The cautionary advice is well taken. We have stressed less on the simplicity of the 3DP design, but emphasized the necessity of the key components (manufacturing facility, know-how, supplies). Overcoming such blockages has been one of the main goals of this project, and with continuous refinement, we have been able to utilize more 3DP and less silicone in the final design, which has been taken up with interest in more needy areas outside of Canada with significant needs. We hope that our elaboration of this point in the revised MS’s discussion section has made the claims of feasibility more realistic.

Response to Reviewer 2’s comments:

1. “The paper presents on the validation approach for reusable respirators that can be used at hospitals and front line workers to access the reliability, reusability of N95 masks with replaceable filters. The paper is timely and should be published as is.”

Thank you. We will expedite this MS’s resubmission process to ensure that important and merited findings are published soon in the face of this ongoing pandemic.

Truly,

William C. K. Ng

MBBS MMed FANZCA FRCPC

Anesthesia and Pain Management, Toronto General Hospital

TGH Research Institute & UHN Advanced Perioperative Imaging Lab

Pediatric Cardiac Anesthesia, SickKids Hospital

Faculty Dept. of Anesthesiology, University of Toronto

M 1.857.330.7399

E William.Ng@uhn.ca or William.Ng@sickkids.ca

Soli Deo Gloria

---

## [Decision Letter · Decision Letter 1]

2 Nov 2020

Subject validation of reusable N95 stop-gap filtering facepiece respirators in COVID-19 pandemic

PONE-D-20-18656R1

Dear Dr. Ng,

We’re pleased to inform you that your manuscript has been judged scientifically suitable for publication and will be formally accepted for publication once it meets all outstanding technical requirements.

Kind regards,

Vahid Serpooshan, PhD

Academic Editor

PLOS ONE

Additional Editor Comments (optional):

Reviewers' comments:

Reviewer's Responses to Questions

**Comments to the Author**

1. If the authors have adequately addressed your comments raised in a previous round of review and you feel that this manuscript is now acceptable for publication, you may indicate that here to bypass the “Comments to the Author” section, enter your conflict of interest statement in the “Confidential to Editor” section, and submit your "Accept" recommendation.

Reviewer #1: All comments have been addressed

2. Is the manuscript technically sound, and do the data support the conclusions?

Reviewer #1: Yes

3. Has the statistical analysis been performed appropriately and rigorously? 

Reviewer #1: Yes

4. Have the authors made all data underlying the findings in their manuscript fully available?

Reviewer #1: Yes

5. Is the manuscript presented in an intelligible fashion and written in standard English?

Reviewer #1: Yes

6. Review Comments to the Author

Reviewer #1: I would caution the authors about assuming that more facial tissue (re. obese and overweight users) would create a better in all, or even most cases. Depending on the unique user, a one-size-fits-all mask approach which seems to be the chosen approach for translation here might not cover a sufficient range of the general population's facial features. As adding 3D facial mapping for each user is unfeasible for a stop-gap design, will add to production steps and time, and raise the already high price per unit of the proposed design here, I would suggest that several sizes of masks be designed in the production stage with industry collaborators, and a small section be added in the manuscript commenting on that.

7. PLOS authors have the option to publish the peer review history of their article (what does this mean?). If published, this will include your full peer review and any attached files.

Reviewer #1: No

---

## [Editor Report · Acceptance letter]

4 Nov 2020

PONE-D-20-18656R1 

Subject validation of reusable N95 stop-gap filtering facepiece respirators in COVID-19 pandemic 

Dear Dr. Ng:

I'm pleased to inform you that your manuscript has been deemed suitable for publication in PLOS ONE. Congratulations! Your manuscript is now with our production department. 

Kind regards, 

on behalf of

Dr. Vahid Serpooshan 

Academic Editor

PLOS ONE